# Diseased Fish Detection in the Underwater Environment Using an Improved YOLOV5 Network for Intensive Aquaculture

**Zhen Wang** [1,2], **Haolu Liu** [3,*] , **Guangyue Zhang** [3], **Xiao Yang** [1,2], **Lingmei Wen** [1,2] and **Wei Zhao** [1,2]

1   Xianning Academy of Agriculture Sciences, Xianning 437100, China
2   Xianning Branch, Hubei Academy of Agricultural Sciences, Xianning 437100, China
3   Nanjing Institute of Agricultural Mechanization, Ministry of Agriculture and Rural Affairs, Nanjing 210049, China
*   Correspondence: liuhaolu@caas.cn; Tel.: +86-173-7277-8342

**Abstract:** In intensive aquaculture, the real-time detection and monitoring of common infectious disease is an important basis for scientific fish epidemic prevention strategies that can effectively reduce fish mortality and economic loss. However, low-quality underwater images and low-identification targets present great challenges to diseased fish detection. To overcome these challenges, this paper proposes a diseased fish detection model, using an improved YOLOV5 network for aquaculture (DFYOLO). The specific implementation methods are as follows: (1) the C3 structure is used instead of the CSPNet structure of the YOLOV5 model to facilitate the industrial deployment of the algorithm; (2) all the $3 \times 3$ convolutional kernels in the backbone network are replaced by a convolutional kernel group consisting of parallel $3 \times 3$, $1 \times 3$ and $3 \times 1$ convolutional kernels; and (3) the convolutional block attention module is added to the YOLOV5 algorithm. Experimental results in a fishing ground showed that the DFYOLO is better than that of the original YOLOV5 network, and the average precision was improved from 94.52% to 99.38% (when the intersection over union is 0.5), for an increase of 4.86%. Therefore, the DFYOLO network can effectively detect diseased fish and is applicable in intensive aquaculture.

**Keywords:** real-time; epidemic prevention; algorithm; target detection; convolution kernel group



## 1. Introduction

China is the largest consumer of freshwater fish in the world [1,2]. Wild-caught freshwater fish make up a large portion of freshwater fish consumption [3]. Overfishing due to consumption of wild-caught freshwater fish destroys biodiversity [4]. In order to reduce the ecological damage caused by fishing, in 2020, the Chinese Ministry of Agriculture and Rural Affairs announced the start of a 10-year fishing ban on the Yangtze River [4]. With the implementation of the fishing ban, China's freshwater fish consumer market is more dependent on artificial culture. During freshwater aquaculture, fish feed or fertilizer is put into the water to increase freshwater fish production, but residual fertilizer, fish feces and other excreta can cause water eutrophication and lead to eco-catastrophe, such as red tides, so this type of freshwater aquaculture was banned in natural waters. In this context, the proportion of intensive aquaculture will further increase [5,6]. One study shows that intensive aquaculture systems will have to predominate [7]. The high-density stocking and feeding method will inevitably face the problem of excessive multiplication of viruses, bacteria and fungi, and the accumulation of nitrogen and phosphorus, especially ammonia and nitrite nitrogen concentrations, making farmed fish more susceptible to various diseases. One study showed that more than half of production losses in aquaculture are caused by diseases [8]. At present, fish disease detection mainly relies on manual methods. However, as light is refracted from the air into water, the human eye observes the health of fish with difficulty, leading to failure of timely drug administration or adjustment of breeding programs, missing the best treatment period, and resulting in huge economic

losses. This is, thus, the technical bottleneck of intensive aquaculture. Therefore, it is important to study automatic fish disease identification and analysis methods.

With the development of deep learning technology, many researchers have proposed to identify animal behavior, and animal and plant body surface characteristics based on video image analysis, so as to identify animals in heat, hunger and disease, and plant maturity and disease, and great progress has been made in recent years. Chen et al. [9] proposed a fish species identification system for fish markets that combines state-of-the-art instance-segmentation methods, with ResNet-based classification. Rauf et al. [10] proposed a deep learning framework based on a CNN approach for fish species recognition, which has been shown to achieve state-of-the-art performance through experimental comparisons with other deep learning frameworks. Qi et al. [11] proposed a novel lightweight convolutional neural network for medicinal chrysanthemum detection. Måløy et al. [12] proposed a two-stream recurrent network (DSRN) to automatically capture the spatio-temporal behavior of salmonids as they swim, and the model achieved a prediction accuracy of 80%. Zhang et al. [13] proposed an automated fish population counting method based on machine vision and a new hybrid deep neural network model to count farmed Atlantic salmon. Labao and Naval [14] proposed a fish detection system consisting of an ensemble of region-based convolutional neural networks, that can detect and count fish objects under various benthic backgrounds and illumination conditions. The rapid development of intelligence has been applied in aquaculture.

In order to reduce the losses caused by fish infectious diseases, the detection method should have a high level of real-time. The You Only Look Once (YOLO) series algorithm, now updated to the fifth generation, is the mainstream single-stage detection algorithm with high detection accuracy and fast detection speed, which is widely used in various target detection tasks. Roy et al. [15] proposed a mango detection framework based on an improved YOLOv4 algorithm, by including DenseNet in the backbone. Fan et al. [16] proposed a real-time apple defects inspection method based on a simplified YOLOV4 algorithm, using channel pruning and layer pruning methods. Qi et al. [17] proposed a highly fused and lightweight deep learning architecture, based on YOLO, for tea chrysanthemum detection. Ge et al. [18] proposed the UW_YOLOv3 lightweight model to solve the problems of calculating energy consumption and storage resource limitations in underwater application scenarios. Cai et al. [19] proposed a fish detection method combining YOLOv3 and MobileNetv1, which was used to detect the number of fish within real breeding farms. Li et al. [20] proposed a real-time fish detection method based on YOLO-V3-Tiny-MobileNet, using YOLO-V3-Tiny as the baseline, and combined with MobileNet, the method can provided timely warning to fishing vessel operators. Hu et al. [21] proposed an uneaten feed pellet detection model, using an improved YOLO-V4 network. Jalal et al. [22] proposed a hybrid solution combining optical flow and Gaussian mixture models with YOLO deep neural network, which achieved 95.47% and 91.2% F-scores for fish detection, while the accuracy of fish species classification was 91.64% and 79.8%, respectively. Wen et al. [23] proposed a lightweight YOLOv4 detection algorithm improved by multi-scale feature fusion for dense silkworm detection. Zhou et al. [24] proposed a marine biometric identification method based on image enhancement and improved YOLOv4 algorithm. Prasetyo et al. [25] fish body part detection method, based on YOLOV4-tiny with a wing convolution layer. Abinaya et al. [26] proposed a YOLOv4 based fish segmentation detection method to estimate the fish biomass in an occulted environment. Wang et al. [27] proposed a neural network based on improved YOLOV5 and SiamRPN++, to detect and track the abnormal behavior of porphyry seabream. YOLO has been used successfully in many applications for target detection in aquaculture. However, the original YOLO algorithm still has limitations for the detection of fish infectious diseases due to the small difference between diseased and normal fish.

To solve the above problems, this paper proposes a YOLOV5m network-based diseased fish detection method for intensive aquaculture. The recognition algorithm replaces all the $3 \times 3$ convolutional kernels in the backbone network with a convolution kernel group,

and adds a convolutional block attention module (CBAM), which effectively improves the detection accuracy of diseased fish. The next sections of this paper are organized as follows: Section 2 introduces the dataset, the improved algorithm and network; Sections 3 and 4 present the results and discussion; and Section 5 is the conclusion.

## 2. Materials and Methods

### 2.1. Dataset

Fish image data were collected in the aquaculture base of Xianning Academy of Agricultural Sciences, Hubei Province. The shooting scene is as shown in Figure 1.

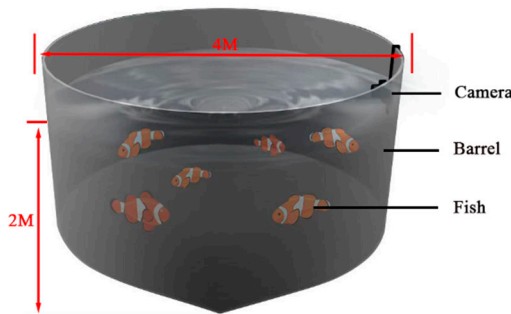

**Figure 1.** The experimental setup of image acquisition.

The acquisition equipment consisted of the GO POR8 motion camera (GO POR, San Mateo, CA, USA), of which the two-dimensional RGB image resolution was 5650 × 4238 pixels, focal length was 2 mm, and exposure time was 1/200 s. In order to reduce the probability of network model overfitting caused by insufficient diversity of training samples, strong light and weak light at acquisition time were distinguished by 500–1000 lux and 50–499 lux of underwater light intensity of the day, respectively. Under strong light and weak light, fish feature images of 600 mm, 1200 mm and 1800 mm depth of water body were collected. The dimensions of the barrel and image acquisition are shown in Figure 1. In order to increase the diversity of the samples, the image samples included two species of fish, red tilapia and micropterus salmoides, including the different conditions of fish density, growth phase, temperature and water quality, as well as light conditions such as light and backlight. Figure 2 shows a group of typical images of fish in complex underwater environments.

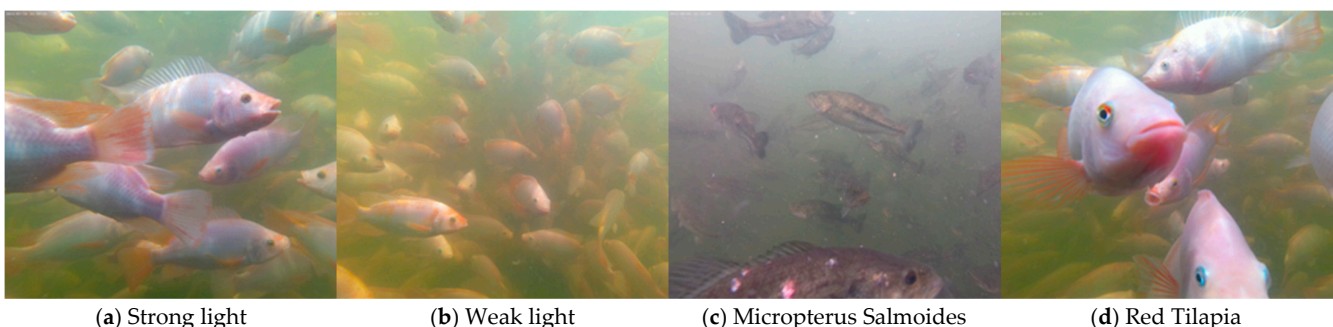

(**a**) Strong light    (**b**) Weak light    (**c**) Micropterus Salmoides    (**d**) Red Tilapia

**Figure 2.** Example images in the underwater environment.

The complex underwater environment of the captive barrel (Fisheries College of Huazhong Agricultural University, Wuhan, China) and the various underwater electronic sensors can interfere with the acquisition and transmission of the images, making them noisy. In order to improve the recognition efficiency of diseased fish, the images need to be processed for noise reduction. Analysis of the image noise shows that the probability density function obeys a Gaussian distribution, and the common processing methods are Gaussian filtering or bilateral filtering. Gaussian filtering has good performance in low-pass

filtering algorithms, but it only takes into account the spatial position of the pixels, and the filtering results in the loss of edge information. The edges in this case are the main areas of color in the image (turbidity of the water, individual body colors of red tilapia, etc.), and bilateral filtering solves this problem by adding an additional weight to the Gaussian filter.

Photographs taken between 1 July and 31 October were manually checked and invalid photographs of poor quality were removed. Including 12,045 photographs of rainbow snapper and 12,317 photographs of micropterus salmoides, 24,362 images were obtained. As the photographs were taken underwater over a certain period of time, there was turbidity in the water as the weather and the environment in the captive barrel changed, a phenomenon that can be likened to taking photographs in the air due to haze, resulting in poor image characteristics and affecting image quality. Therefore, it is necessary to use a dark channel defogging method on the collected samples, to saturate the color and increase the brightness of the photographs to facilitate the identification of lesions on the surface of the fish. The expression for haze removal using dark channel [28] is shown in Equation (1).

$$J_{(x)} = \frac{I_{(x)} - A}{t_{(x)}} + A \tag{1}$$

when the transmittance, $t_{(x)}$, is too small, the value of $J_{(x)}$ will be large and a threshold $t_{(0)}$ is set to limit $J_{(x)}$, typically taking the value 0.1. Then, there is the Equation (2), shown below.

$$J_{(x)} = \frac{I_{(x)} - A}{max\left(t_{(x)}, t_0\right)} + A \, , \tag{2}$$

where $x$ is the input pixels, $J_{(x)}$ is the recovered haze-free image, $A$ is the atmospheric illumination (maximum pixel value), $I_{(x)}$ is the existing images (to be foggy) and $t_{(x)}$ is the medium transmission rate.

The pathological analysis showed that the main disease of red tilapia was ocular streptococcal infection, which caused congestion, protrusion and ulceration of pus in the eyes of red tilapia. The main disease of micropterus salmoides was Nocardia infection, which caused obvious redness, inflammation, ulceration and necrosis on the surface of the body of both fish. A comparison of the symptoms of the two species is shown in Figure 3.

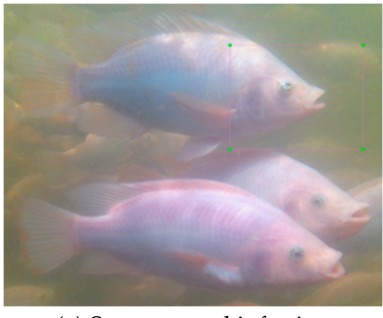 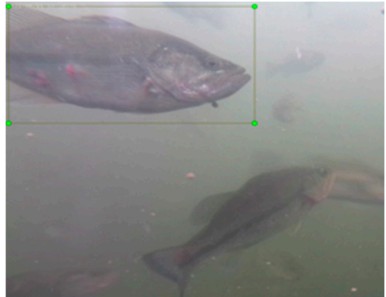

(**a**) Streptococcal infection　　　　　　(**b**) Nocardia infection

**Figure 3.** Typical images of fish infection. Note: The infected fish in the figure are inside the Ground truth box, and normal fish are found outside the Ground truth box.

To further clarify the trend of fish infection with culture time, 25 images of red tilapia and micropterus salmoides were selected for tagging each day, and a total of 6000 images were tagged, with 50% of each of the two species. The labelled images were divided into a train set and a validation set according to 80% and 20%, and another 1000 images were selected from the images taken as a test set, with 50% of each species. The number of disease and illness levels for each fish species in the captive barrel was counted, as shown in Table 1. Three thousand images of red tilapia were marked, of which 785 had eye infections and 213 died, while the remaining healthy fish were marked as normal. Of the 3000 images of

micropterus salmoides, a total of 2080 fish had surface decay and the remaining healthy fish were marked as normal.

**Table 1.** Various state statistics of the fish in the dataset.

| Collection Name | Red Tilapia | Micropterus Salmoides |
|:---:|:---:|:---:|
| Eye infections | 785 | 0 |
| Surface decay | 0 | 2080 |
| Died | 213 | 0 |
| Normal | 2002 | 920 |
| Summation | 3000 | 3000 |

*2.2. The Diseased Fish Identification Algorithms*

The YOLOV5 network is a current excellent one-stage target detection algorithm, of which the structure is divided into three parts: backbone, neck and head. The main component of YOLOV5 includes Focus, SPP and PANet, where Focus improves the receptive field through slice and convolution operations, SPP separates the important features of the context through max pool and convolution operations, and PANet obtains the strong feature map by integrating different level features.

There are four types of YOLOV5: V5s, V5m, V5l and V5x. Taking speed and accuracy into account, this paper proposes a diseased fish detection model based on YOLOV5m (DFYOLO), the network structure is shown in Figure 4.

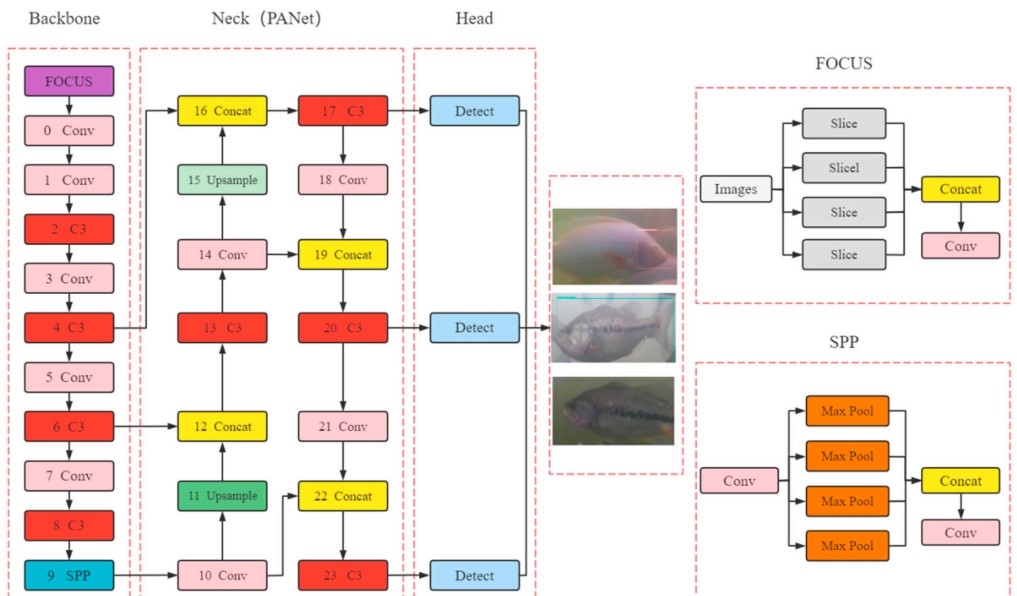

**Figure 4.** Algorithm structure and workflow of DFYOLO. Note: Focus is the slice operation, and it can double down sample feature graph without information loss; SPP is a spatial pyramid pool structure, and it effectively avoids the problems of image distortion caused by image region cropping and scaling operation.

The main model improvements of this study are as follows:

1. The implementation of lightweight computing is key to the industrial deployment of algorithms. In order to achieve lightweight computing, the C3 component is used instead of the CSPNet [29] component of the YOLOV5 model in this study. As shown in Figure 5, compared to the CSPNet component, the C3 component removes one convolution operation after the skip connect, and splices directly with another branch of the input image after one convolution operation. The C3 component has a similar effect to the CSPNet component, but with a simpler structure.

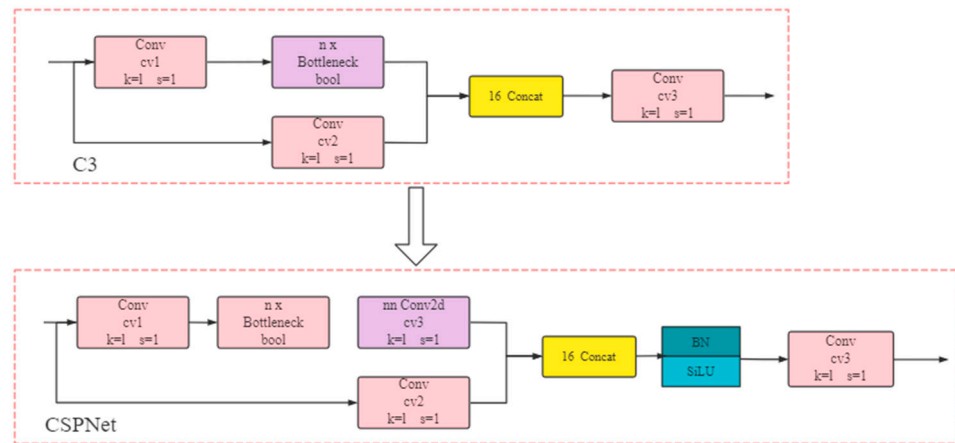

**Figure 5.** Schematic diagram of the component replacement. Note: The C3 component replaces the CSPNet component.

2. As fish activity photos are taken underwater, the scenes in the photos often change with water quality, lighting and fish conditions, resulting in varying degrees of variation and obscuration of fish features, and the original YOLOV5 backbone network is unable to extract clear features. In order to reduce the impact of underwater complexities, this study improves the generalization ability of the network by expanding the training samples and replacing all $3 \times 3$ convolutional kernels in the backbone network with convolutional kernel group (Conv KG), to enhance the network's ability to extract features from the photographed fish. Conv KG consists of three parallel $3 \times 3$, $1 \times 3$ and $3 \times 1$ convolutional kernels, which convolve the input image in the same steps to produce feature maps of the same size and number of channels, respectively, and the corresponding feature maps are summed to obtain the output features, as shown in Figure 6. The three parallel convolutional kernels enhance the network's ability to extract fish surface features.

$$P * K(1) + P * K(2) + P * K(3) = P * (K(1) + K(2) + K(3)) = P * K, \tag{3}$$

where $P$ is the input image, $K(1)$, $K(2)$ and $K(3)$ are the $3 \times 3$, $1 \times 3$ and $3 \times 1$ convolution kernels, $K$ is the equivalent convolution kernel, $*$ is the convolution operation.

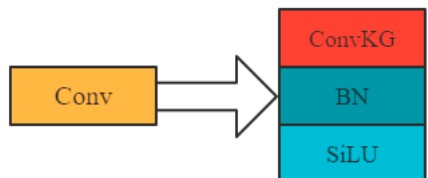

**Figure 6.** Improved convolution module.

Therefore, the three parallel convolutional kernels are equivalent to a new $3 \times 3$ convolutional kernel with different weights. Compared to the $3 \times 3$ kernels before the replacement, the trained equivalent convolutional kernel can enhance the extraction of fish surface features, without increasing the computational effort in the train set.

3. Due to the variable underwater environment and the large variation in the number of fish in fish photos, which interferes with the original YOLOV5 detection algorithm, this study adds the attention mechanism module, convolutional block attention module (CBAM), to the YOLOV5 network (Figure 7). CBAM is a simple, but effective feed-forward convolutional neural network attention module, which combines the channel attention module (CAM) and spatial attention module (SAM) [30]. Given an intermediate feature map, our module sequentially inferred the attention of an image along two independent dimensions, channel and space, and then multiplied

the attention map by the input feature map for adaptive feature refinement. Because CBAM is a lightweight, general-purpose module, it can be seamlessly integrated into any convolutional neural network architecture with negligible overhead, and can be trained end-to-end with the underlying convolutional neural network.

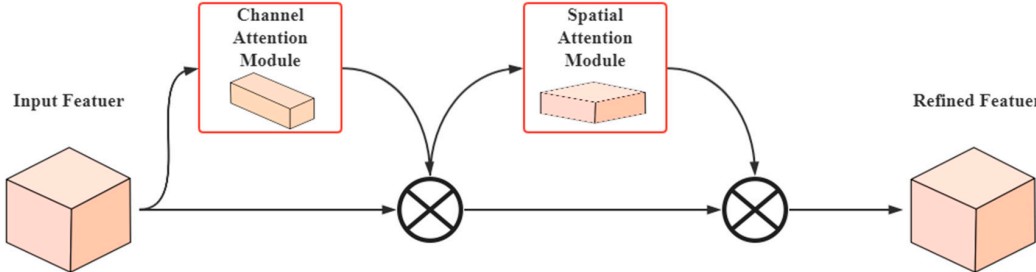

**Figure 7.** Convolutional block attention module.

The overall CBAM process is shown in Equations (4) and (5):

$$F' = M_C(F) \otimes F \tag{4}$$

$$F'' = M_S(F') \otimes F', \tag{5}$$

where $F \in R^{C \times H \times W}$ is the input feature map, $M_C \in R^{C \times 1 \times 1}$ is the CAM weight data, which is $1 \times 1 \times C$, $F'$ is the CAM output, $M_S \in R^{1 \times H \times W}$ is the spatial attention module weight data, which is $2 \times H \times W$ and $F''$ is the CBAM output.

The channel attention module mainly detects the contour characteristics of the fish and obtains the main contents of the detection target. Its calculation method is as follows:

$$M_C(F) = \sigma\left(W_1\left(W_0\left(F_{avg}^c\right)\right) + W_1(W_0(F_{max}^c))\right) \tag{6}$$

In the formula, $\sigma$ represents the Sigmoid function, $W_0 \in R^{c/r \times c}$, $W_1 \in R^{c \times c/r}$, two inputs share the weights $W_0$ and $W_1$, the ReLU activation function is followed by the $W_0$, $F_{avg}^c$, $F_{max}^c$, representing features generated using average-pooled and max-pooled, $H$ is the height, $W$ is the width, $C$ is the number of channels, and r is the rate of reduction.

The introduction of the spatial attention module is to further improve the detection accuracy of the target fish, and accurately locate the position of target fish. The calculation method is as follows:

$$M_S(F) = \sigma\left(f^{7 \times 7}\left(\left[F_{avg}^s; F_{max}^s\right]\right)\right), \tag{7}$$

where $F_{avg}^s$ and $F_{max}^s$ represent the features generated using average-pooled and max-pooled and $f^{7 \times 7}$ represents a convolution operation with the filter size of $7 \times 7$.

### 2.3. Experimental Environment

In this study, the Darknet framework was used to improve the YOLOV5 network. The experimental environment is shown in Table 2.

**Table 2.** The experimental environment.

| Configuration | Parameter |
| --- | --- |
| CPU | Intel Core i7-9700 K |
| GPU | Nvidia GeForce RTX 3080 Ti $\times$ 2 |
| Operating system | Windows 10 |
| Accelerated environment | CUDA10.2 CUDNN7.6.5 |
| Development environment | Visual Studio 2020 |
| Library | Opencv3.4.0 |

Note: CPU manufactured by Intel Corporation, Santa Clara, CA, USA; GPU manufactured by Nvidia Corporation, Santa Clara, CA, USA; the operating system is manufactured by Microsoft Corporation, Albuquerque, NM, USA.

*2.4. Performance Evaluation Metrics*

In this study, precision (P), recall (R) and mean average precision (mAP) are used as evaluation indicators for DFYOLO. Precision is the probability of an actual diseased fish being predicted in all samples predicted to be diseased, which represents the precision of the predicted outcome for positive samples. Recall is the probability of being predicted as a diseased fish among the actual diseased fish samples, and it represents the overall prediction accuracy. It is calculated as shown by Equations (8) and (9).

$$precision = \frac{TP}{TP + FP} \tag{8}$$

$$recall = \frac{TP}{TP + FN}, \tag{9}$$

where *TP* (true positives) is a sample that is correctly identified as abnormal behavior; *FN* (false negatives) is a sample that is mistaken as a background; *TN* (true negatives) is a sample that is correctly identified as a background; *FP* (false positives) is a sample that is misidentified as abnormal behavior.

Average precision (*AP*) refers to the area under the P–R curve, for which the calculation formula is shown by Equation (10). AP50 is the mean of precision under different recall values when IOU (intersection over union) = 0.5. AP50:95 is the mean of the ten values of AP50, AP55 . . . , AP90, AP95, for which the calculation formula is shown by Equation (11). The mAP refers to the average of the *AP* values of the two species, micropterus salmoides (*MS*) and red tilapia (*RT*). The calculation formula is shown by Equation (12).

$$AP = \int_0^1 P(R)dR \tag{10}$$

$$AP_{50:95} = \frac{1}{10}(AP_{50} + AP_{55} + \ldots + AP_{90} + AP_{95}) \tag{11}$$

$$mAP = \frac{1}{2}(AP_{MS} + AP_{RT}) \tag{12}$$

## 3. Results

*3.1. Training Result*

The training results of DFYOLO are shown in Figure 8. After 475 times epoch, the loss value dropped to 0.027. The accuracy of DFYOLO was 99.75%, the recall was 99.31% and the mAP50 was 99.38%. From the training results, the model has convergence with high accuracy and check-all rates, as well as low loss value.

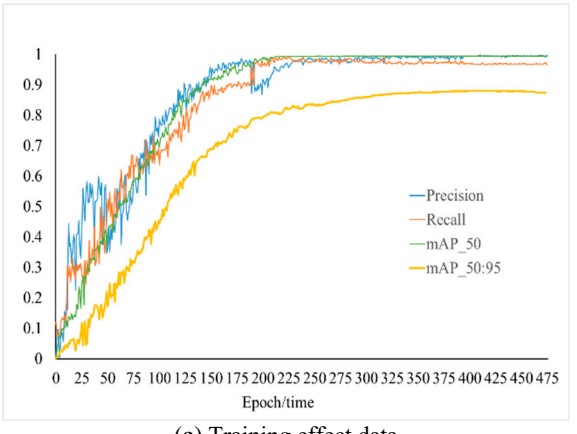

(**a**) Training effect data      (**b**)Training loss date

**Figure 8.** Training results.

### 3.2. Identification Results under Various Varieties

The images of red tilapia and micropterus salmoides were randomly collected at the same light intensity and water depth, and entered into DFYOLO. The monitoring results are shown in Figure 9. When red tilapia were close to the camera and the sides and their heads were fully displayed, no missed detections occurred; the micropterus salmoides that had their side torsos fully displayed in the image were able to be detected in their entirety, with no missed detections.

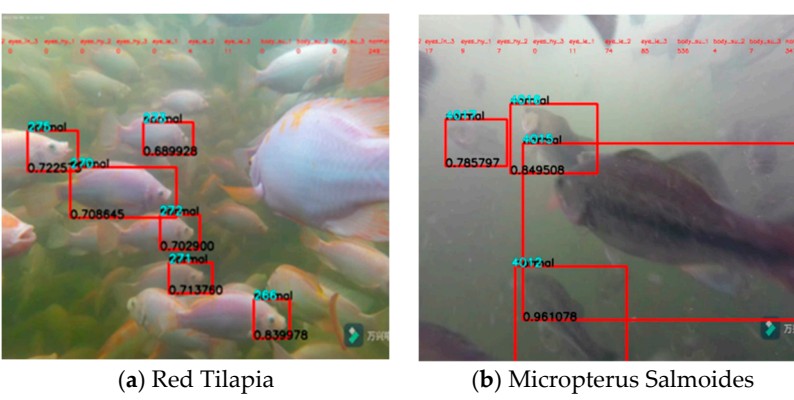

(**a**) Red Tilapia    (**b**) Micropterus Salmoides

**Figure 9.** Monitoring effect of different varieties.

### 3.3. Identification Results under Various Environments

The images of red tilapia and micropterus salmoides were randomly selected at 500–1000 lux (strong light) and 50–499 lux (weak light), respectively, and input into DFYOLO. The monitoring results are shown in Figure 10, and there were no missed detections, indicating that DFYOLO can adapt to the different lighting conditions common in the breeding process.

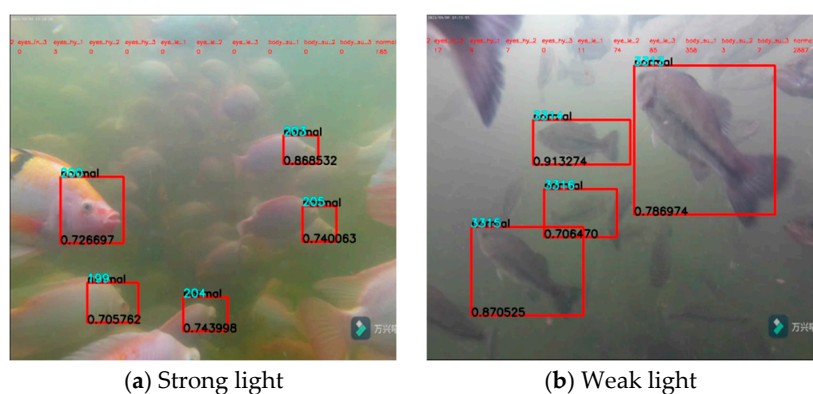

(**a**) Strong light    (**b**) Weak light

**Figure 10.** Monitoring effect at different light conditions.

The monitoring results for different water depths are shown in Figure 11. The images taken at 600 mm depth had the best detection results, because there were fewer fish in the upper layer and fewer fish adhered to the images; the images taken at 1200 mm depth had no missed detection, which was slightly less effective than those taken at 600 mm depth; the images taken at 1800 mm depth had missed detection of individual fish, as shown by the pink dashed circle in Figure 11, indicating that DFYOLO had better detection at different shooting depths. This shows that DFYOLO has a good monitoring effect at different shooting heights.

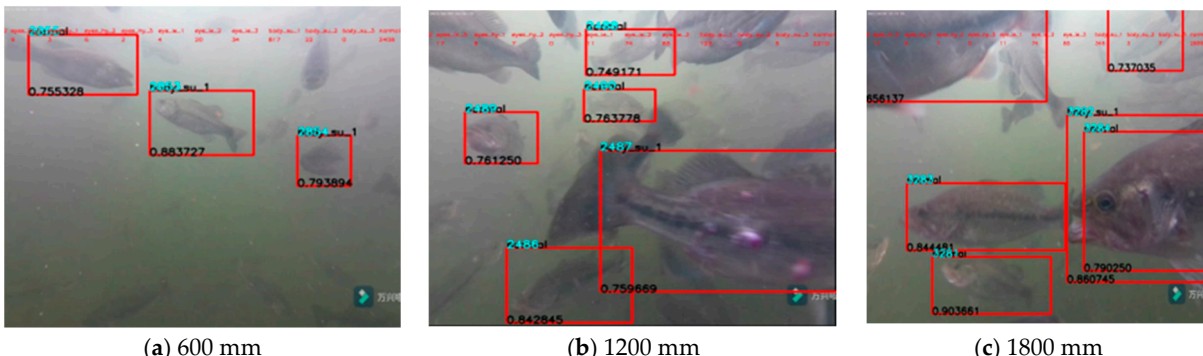

(**a**) 600 mm         (**b**) 1200 mm         (**c**) 1800 mm

**Figure 11.** Monitoring effect at different water depth.

### 3.4. Contrast to Mainstream Target Detection Networks

In order to verify the detection performance of the improved algorithm in this paper, the DFYOLO proposed in this study was compared with the mainstream target detection models SSD, Faster-R CNN, YOLOv3 and YOLOv4, and the five metrics of Occupy Memory, FPS, Recall, Precision and mAP50, were used to evaluate and compare the mainstream detection algorithms. The comparison experimental parameters are shown in Table 3.

**Table 3.** Parameter settings for different detection networks.

| Method | Batch Size | Learning Rate | Epoch | Momentum | Weight Decay |
|---|---|---|---|---|---|
| SSD | 16 | 0.001 | 475 | 0.9 | 0.0005 |
| Faster-RCNN | 16 | 0.001 | 475 | 0.9 | 0.0005 |
| YOLOv3 | 16 | 0.001 | 475 | 0.9 | 0.0005 |
| YOLOv4 | 16 | 0.001 | 475 | 0.9 | 0.0005 |
| YOLOV5 | 16 | 0.01 | 475 | 0.9 | 0.0005 |
| DFYOLO | 16 | 0.01 | 475 | 0.9 | 0.0005 |

The analysis in Table 4 shows that when the IOU = 0.5, the recall of the DFYOLO algorithm was 99.31%, which is 5.55% higher than the original YOLOV5 model; the precision was 99.75%, which is 5.39% higher than the original YOLOV5 model; the mAP50 was 99.38%, which is 5.86% higher than the original YOLOV5 algorithm; the mAP50:95 was 88.09%, which is 9.56% higher than the original YOLOV5 algorithm; the model occupied 0.9 MB less memory; while ensuring high accuracy detection, the FPS of the model did not show a significant decrease.

**Table 4.** The detection results of different detection networks.

| Method | Occupy Memory/MB | FPS | Recall/% | Precision/% | mAP50/% | mAP50:95/% |
|---|---|---|---|---|---|---|
| SSD | 63.2 | 69.73 | 73.01 | 76.81 | 68.24 | 42.03 |
| Faster-RCNN | 77.8 | 57.97 | 83.28 | 86.18 | 79.87 | 51.14 |
| YOLOv3 | 62.1 | 69.44 | 86.12 | 82.38 | 90.99 | 41.10 |
| YOLOv4 | 226 | 78.24 | 93.84 | 93.58 | 93.39 | 59.53 |
| YOLOV5 | 14.5 | 96.43 | 93.76 | 94.36 | 94.52 | 78.53 |
| DFYOLO | 13.6 | 93.21 | 99.31 | 99.75 | 99.38 | 88.09 |

### 3.5. Ablation Experiment

An ablation experiment is a commonly used experimental method in deep learning, which proves the necessity of this module by removing the effect of single- or multiple-improved methods. If the results of an ablation experiment are worse or the performance is significantly reduced, the improved method is reasonable.

The results of the ablation experiment are shown in Table 5, where it can be seen that bilateral filtering and haze removal improved the quality of underwater photos, and slightly improved the model detection. Image transformation can reduce the computation time of the algorithm, Conv KG enhances the feature extraction of the fish surface, CBAM focuses on "what the fish is?" and "where the fish are? is?". The above methods improved the detection speed, precision recall and mAP of the model.

**Table 5.** The results of an ablation test.

| Baseline Network | Bilateral Filtering | Haze Removal | Image Transformation | Conv KG | CBAM | Recall/% | Precision/% | mAP50/% | mAP50:95/% |
|---|---|---|---|---|---|---|---|---|---|
| YOLOV5 | | | | | | 93.76 | 94.36 | 94.52 | 78.53 |
| YOLOV5 | √ | | | | | 95.81 | 94.40 | 94.58 | 79.28 |
| YOLOV5 | √ | √ | | | | 95.85 | 94.86 | 96.02 | 82.12 |
| YOLOV5 | √ | √ | √ | | | 95.75 | 93.42 | 95.41 | 82.07 |
| YOLOV5 | √ | √ | | √ | | 97.21 | 98.24 | 98.35 | 87.98 |
| YOLOV5 | √ | √ | √ | √ | | 97.18 | 98.85 | 98.16 | 86.06 |
| YOLOV5 | √ | √ | √ | √ | √ | 99.31 | 99.75 | 99.38 | 88.09 |

*3.6. Comparison of before and after Improvement*

Randomly selected photographs of red tilapia and micropterus salmoides were entered into the original YOLOV5 and DFYOLO models, as shown in Figure 12. The thin line part is the recognition result of the original model, the thick line is the recognition result of the DFYOLO, and the fish missed by the original model are only marked by the dashed line. By comparing the images, the DFYOLO had a higher detection efficiency, more accurate detection parts and a lower missed detection rate.

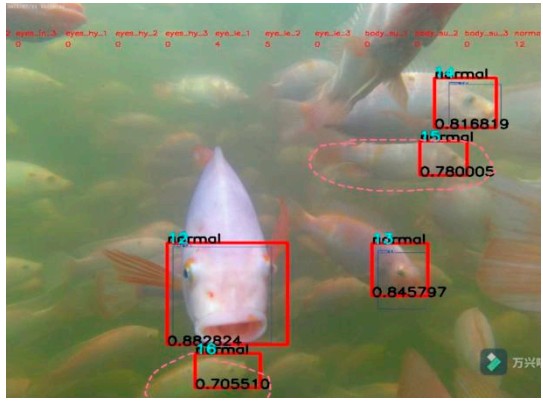 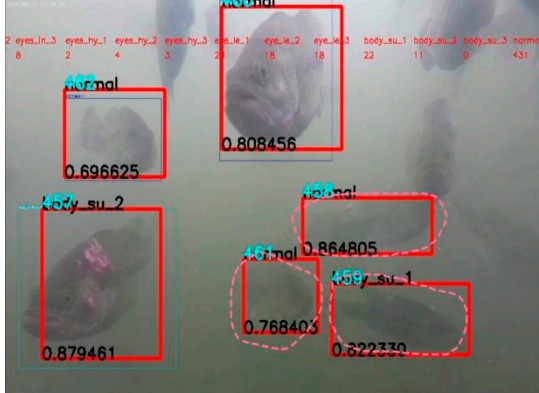

(**a**) Red Tilapia　　　　　　　　　　　　　　　　(**b**) Micropterus Salmoides

**Figure 12.** Comparison of the test image results.

## 4. Discussion

*4.1. Robustness of the Method*

Although our work is an exploratory experiment, it aims to explore whether recognition algorithms can be used for disease recognition in fish. Nevertheless, we simulated the actual production situation to improve the robustness of our method. We chose to use a large intensive culture barrel for our experiment. First, we took time series of photographs of fish at different stages of growth. Secondly, we also selected two typical varieties (Figure 9), different light conditions (Figure 10) and different water depths (Figure 11), which were as close to the real environment as possible. In addition, the experimental design added some algae, so that we could see that all the photos were yellowish-green. Our models still work well in this underwater environment.

*4.2. Comparison with Other Methods*

At present, there are also many underwater fish tracking and identification methods used [31,32]. They generally use RCNN or CNN, which has a high recognition accuracy and a low technical threshold. However, due to its structure, the recognition speed is not fast enough. Intensive farming requires relatively fast recognition speed, because more fish targets appear per unit image area. Therefore, this study preferentially selects YOLO as the basic algorithm. According to the data in Table 4, YOLOv5 has an obvious speed advantage compared with other algorithms in actual use. After some improvements, the DFYOLO sacrifices some of its speed advantage to reduce memory usage, but still achieves 93.21FPS.

*4.3. Contribution to the Detection of Fish Diseases*

In recent years, with the rapid development of artificial intelligence, the application of computer vision model in fishery industry is more and more popular. This study is concerned with the use of computer vision technology for real-time detection of fish body surface to estimate the disease and degree of fish, which can improve the management efficiency of freshwater fish culture. The difference between underwater fish disease recognition and common target recognition is that the features of fish body surface are more homogenous and smaller than those of common targets. On the other hand, the underwater shooting environment is much more complex than that in the air. In addition, the most important point is that fish have a large degree of overlap in space, so it is easy to introduce redundant features and affect the accuracy of body surface disease recognition. In this study, the image noise reduction is carried out by the method of bilateral filtering, which reduces the interference of other electrical equipment to the underwater camera. The application of haze removal using dark channel reduces the influence of turbid water on the image quality, and C3 replacing the original CPSNet reduces the memory occupied by the model, while CABM (convolutional block attention module) is introduced to avoid the influence of a large amount of redundant information on fish recognition. According to the data in Table 5, compared with the original algorithm before improvement (YOLO v5), our model has great advantages in speed and accuracy, achieving a real-time detection speed and accuracy of 99.75%.

**5. Conclusions**

In order to detect fish health in real time, this paper proposes a DFYOLO network based on YOLOV5m, to detect the proportion of diseased fish in the shoal of fish.

The captive barrel has an automatic aerator, sewage discharge and other electrical equipment, generating the electromagnetic interference in the process of camera shooting. In this study, the form of bilateral filtering is adopted to reduce the noise of the images taken. In addition, intensive culture will lead to turbidity of water and unclear surface features of fish body. If the unprocessed image is directly input into the algorithm, the recognition effect will be greatly reduced. The addition of haze removal using dark channel can effectively reduce the influence of water turbidity on photo quality. C3 replaces CSPNet, which makes the whole algorithm structure simple and efficient. The introduction of Conv KG reduces the memory of the whole algorithm. The application of CABM makes the whole algorithm combined-target detection and feature recognition, and achieves good results in fish detection and body surface feature recognition. The experimental results showed that DFYOLO not only achieves the highest mAP50 (99.38%), which is 4.86% higher than YOLOv5m, but also keeps a high inference speed (93.21FPS) and occupies less memory (13.6 MB). This method will provide a theoretical basis for the development of intelligent fishery.

Although this study has realized the identification and statistics of fish disease, the proposed method still has some limitations.

1. The algorithm used in this paper is a target recognition algorithm. It does not actually support multi-target dynamic tracking. As a result, it is unable to track fish move-

ments, and the information collected is too simple to provide a comprehensive picture of their health.

2. The dataset used in this study for two common fish diseases is not sufficient for practical application. More species of fish disease behavior should be collected to supplement the dataset.

3. In future work, we will use stereo and multispectral cameras to capture images, to reduce the impact of illumination, water quality and refraction on image quality. In addition, we have found a number of methods regarding geometric features that can be used to describe the size, weight and swimming speed of the fish, to construct a more accurate estimation model of the health of the fish.

**Author Contributions:** Conceptualization, Z.W., H.L. and G.Z.; methodology, X.Y. and L.W.; software, Z.W.; validation, W.Z.; writing—original draft preparation, Z.W. writing—review and editing, H.L. All authors have read and agreed to the published version of the manuscript.

**Funding:** This research was financially supported by the Hubei science and technology service fishery industry chain "515" action, Agricultural Science and Technology Innovation Program of Chinese Academy of Agricultural Sciences (ASTIP, CAAS) and Xianning Academy of Agricultural Sciences subject group leader responsibility system project (XNNK20210602).

**Institutional Review Board Statement:** Not applicable.

**Informed Consent Statement:** Not applicable.

**Data Availability Statement:** Not applicable.

**Acknowledgments:** We greatly appreciate the careful and precise reviews by the anonymous reviewers and editors.

**Conflicts of Interest:** The authors declare no conflict of interest.

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
