# Peer review of "Diseased Fish Detection in the Underwater Environment Using an Improved YOLOV5 Network for Intensive Aquaculture"

_fishes, doi:10.3390/fishes8030169_

Round 1

Reviewer 1 Report

I found this manuscript quite interesting, as a first experimental trial that will fix help in the future fast disease detection in aquaculture facilities.

Keywords: Please avoid the use of words already reported in the Title, try to replace them with some related ones.

Line 171: Table 1, with relative data, was missing.

Line 250: I think this should be Table 2. The same for the further Tables 3-5.

References: double check the style.

Best regards

The Reviewer

Reviewer 2 Report

I would like to congratulate the authors because in my opinion, the work is innovative and corresponds to the topics of this journal.

I would like to make some considerations of form and substance.

- The number of significant figures in the expression of results, such as precision expressed as a percentage, must be consistent and appropriate.

- It is not convenient to repeat words from the title in the keywords.

- It is appropriate to indicate the manufacturer, city and country of the equipment used in the work.

- Figures and Tables must appear immediately after being named in the text.

- Latin names must be in italics.

- It is advisable to indicate the design of the experiment.

- In my opinion, the titles of figures 4 and 5 should be more explicit.

- It is necessary to indicate in the text some figures and equations.

- Sections 3.1 and 3.2 should go to materials and methods.

- There is no discussion of the results.

- The conclusions do not seem to be based on the discussion of the results. Rather, they are a way of expressing the results.

Round 2

Reviewer 2 Report

The changes made by the authors do not correct the deficiencies found in the first review

Round 3

Reviewer 2 Report

The manuscript has improved sufficiently to be accepted for publication. Thank you very much for the effort